# A Molecular Binuclear Nickel (II) Schiff Base Complex for Efficient HER Electrocatalysis

Kian Shamskhou [iD], Houssein Awada [iD], Farzaneh Yari, Abdalaziz Aljabour [iD] and Wolfgang Schöfberger *[iD]

Institute of Organic Chemistry, Laboratory for Sustainable Chemistry and Catalysis (LSusCat), Johannes Kepler University Linz (JKU), Altenberger Straße 69, 4040 Linz, Austria; kishamsk@gmail.com (K.S.); houssein.awada@jku.at (H.A.); farzaneh.yari@jku.at (F.Y.); aziz.jabour@gmail.com (A.A.)
* Correspondence: wolfgang.schoefberger@jku.at; Tel.: +43-732-2468-5410

**Abstract:** The hydrogen evolution reaction (HER) has emerged as a focal point in the realm of sustainable energy generation, offering the potential to produce clean hydrogen gas ($H_2$) devoid of pollutants. The pursuit of stable HER electrocatalysts that can reduce our reliance on precious platinum, while still maintaining a high level of catalytic efficiency, presents a significant and ongoing challenge. In this study, we introduce the utilization of a binuclear nickel (II) Schiff base complex known as $[Ni]_2[L]_2$ **2** for the purpose of HER electrocatalysis. The rational design of this electrocatalyst has yielded optimal HER performance, wherein the strategic placement of electronegative heteroatoms in proximity to the metal centers serves to enhance proton affinity. Consequently, this catalyst manifests outstanding HER activity, characterized by a nearly 100% faradaic efficiency (FE) at an overpotential potential of $-0.4$ V versus the reverse hydrogen electrode (RHE), sustained catalytic activity over an extended 80 h electrolysis period, and a commendable turnover number (TON) of 0.0006 $s^{-1}$.

**Keywords:** nickel; electrolysis; catalysis; binuclear complex; hydrogen evolution





## 1. Introduction

The growing climate and energy crises emphasize the necessity to reduce modern reliance on fossil fuels and rather focus efforts on the development of sustainable energy generation methodology [1]. In light of this, electrocatalytic technologies emerge as highly coveted tools for their remarkable capacity to orchestrate intricate chemical transformations, thereby enabling the conversion of readily available feedstocks into energy-rich products [2]. Of these chemical pathways, electrochemical water splitting through the hydrogen evolution reaction (HER) is of particular interest to cleanly produce $H_2$ gas. The allure of $H_2$ gas as a prospective energy carrier is steadily growing, owing to its exceptional energy density and the absence of noxious byproducts during combustion [3]. A critical limitation of $H_2$ gas generation is due to high production costs, as earth-abundant transition metal catalysts struggle to display activity rivaling costly platinum-containing materials [4,5]. Bridging this performance gap between rare-earth and earth-abundant transition metal catalysts is a pivotal stride towards realizing large-scale, industrially viable production of $H_2$ gas for the burgeoning renewable energy sector.

To approach the high electrochemical performance of platinum, an informed and rational design of earth-abundant transition metal catalysts is required. Electrocatalytic transition metal complexes using cobalt, copper, molybdenum, or nickel have all shown promise in their ability to generate $H_2$ gas at desirable working potentials [6–9]. Nickel shows particular promise for HER catalysis as DFT calculations reveal its free energy of hydrogen absorption ($\Delta G_H$) to be optimal out of all non-noble transition metals [10]. These calculations are further supported by previous work from Djing et al. as the use nickel nanoparticles acquires current densities of 10 mA $cm^{-2}$ with a low overpotential of 98 mV [11]. However, while nickel nanoparticles exhibit near-platinum-like performance, challenges arise in terms of long-term stability. The formation of nickel hydride species

during catalysis can diminish the overall catalytic lifespan of the system. To circumvent this limitation, recent advancements in nickel HER catalysis have delved into the realm of molecular design, where carefully crafted ligands are introduced to yield organometallic electrocatalysts of heightened stability and reactivity. This innovative approach holds promise in mitigating the durability concerns associated with nickel-based HER catalysts, paving the way for sustainable and economically viable hydrogen production technologies.

A shared feature among previous works utilizing molecular catalysts is the tuning of electronic properties via ancillary ligands. Providing electron-deficient heteroatoms (O, N, and S) near metal centers through redox-active ligands can promote the absorption of protons to the heteroatom sites, leading to an increase in surface area saturation of hydrogen [12]. In the underlying work, we present a novel binuclear nickel (II) HER electrocatalyst complexed from a Schiff base ligand. This electrocatalyst features two quaternary nickel centers surrounded by electronegative oxygen and nitrogen heteroatoms. The catalyst design enhances HER performance, reaching nearly 100% Faradaic efficiency with an overpotential of $-400$ mV in 0.5 M sulfuric acid solution.

## 2. Results and Discussion

Synthesis of the $[Ni]_2[L]_2$ complex is both cheap and facile (Scheme 1). Preparation of the Schiff base ligand **1** originates from commercially available hydroquinone. Hydroquinone was treated with 1-bromooctane in alkaline conditions to obtain the 4-(octyloxy)phenol. Subsequent reaction with $MgCl_2$, paraformaldehyde, and triethylamine in THF at 70 °C afforded the 2-hydroxy-5-(octyloxy)benzaldehyde in excellent yields. Then, the 4-(octyloxy)phenol was nitrated in benzene and 56% aqueous nitric acid forming 2-nitro-4-(octyloxy)phenol in 54% yield. Hydrogenation of 2-nitro-4-(octyloxy)phenol on Pd/C gave 2-amino-4-(octyloxy)phenol in 90% yield. The synthesis of (Z)-2-((2-hydroxy-5-(octyloxy)benzylidene)amino)-4-(octyloxy)phenol was then performed with the two precursors, 2-amino-4-(octyloxy)phenol and 2-hydroxy-5-(octyloxy)benzaldehyde, in dry EtOH at 80 °C. The Schiff base ligand **1** was obtained in 70% yield.

**Scheme 1.** Synthesis procedure of binuclear $[Ni]_2[L]_2$ catalyst **2** for hydrogen evolution reaction (HER) from the octyloxy Schiff base ligand **1** precursor.

The subsequent metalation of ligand **1** with nickel (II) acetate in ethanol at reflux temperature proceeds easily to afford the $[Ni]_2[L]_2$ complex. Solid-state $^{13}C$ NMR spectroscopy,

MALDI-TOF mass spectrometry, and CHN elemental analysis confirm the proposed composition of [Ni]$_2$[L]$_2$. Solid-state $^{13}$C-NMR confirms the presence of alkyl chain carbons (δ = 33.18–16.14) and aromatic carbons (δ = 113.35–109.45) (Figure 1a). The MALDI-TOF mass spectrum shows the [M + H$^+$]$^+$ at m/z = 1051.165. Consistent with the existing literature and supported by our Density Functional Theory (DFT) and Time-Dependent DFT (TD-DFT) calculations, we observed a noticeable shift in the absorption band within the infrared (IR) spectrum. Specifically, the band at 1622 cm$^{-1}$, which is indicative of the ν (C=N) vibration in the Schiff base ligand, shifted to 1600 cm$^{-1}$ following the formation of the metal complex (Figures S8 and S9) [13]. The utility of this synthetic pathway provides tunability for ligand side chains, as evident in the previous literature using Schiff base ligands for complexation [14]. The inherent nature of the catalyst structure leaves it prone to π–π stacking interactions, limiting solubility [15] (Figure 2a). For this reason, [Ni]$_2$[L]$_2$ makes for an excellent candidate for heterogenous electrocatalysis in aqueous mediums, lowering catalyst loading through adhesion to carbon paper electrodes.

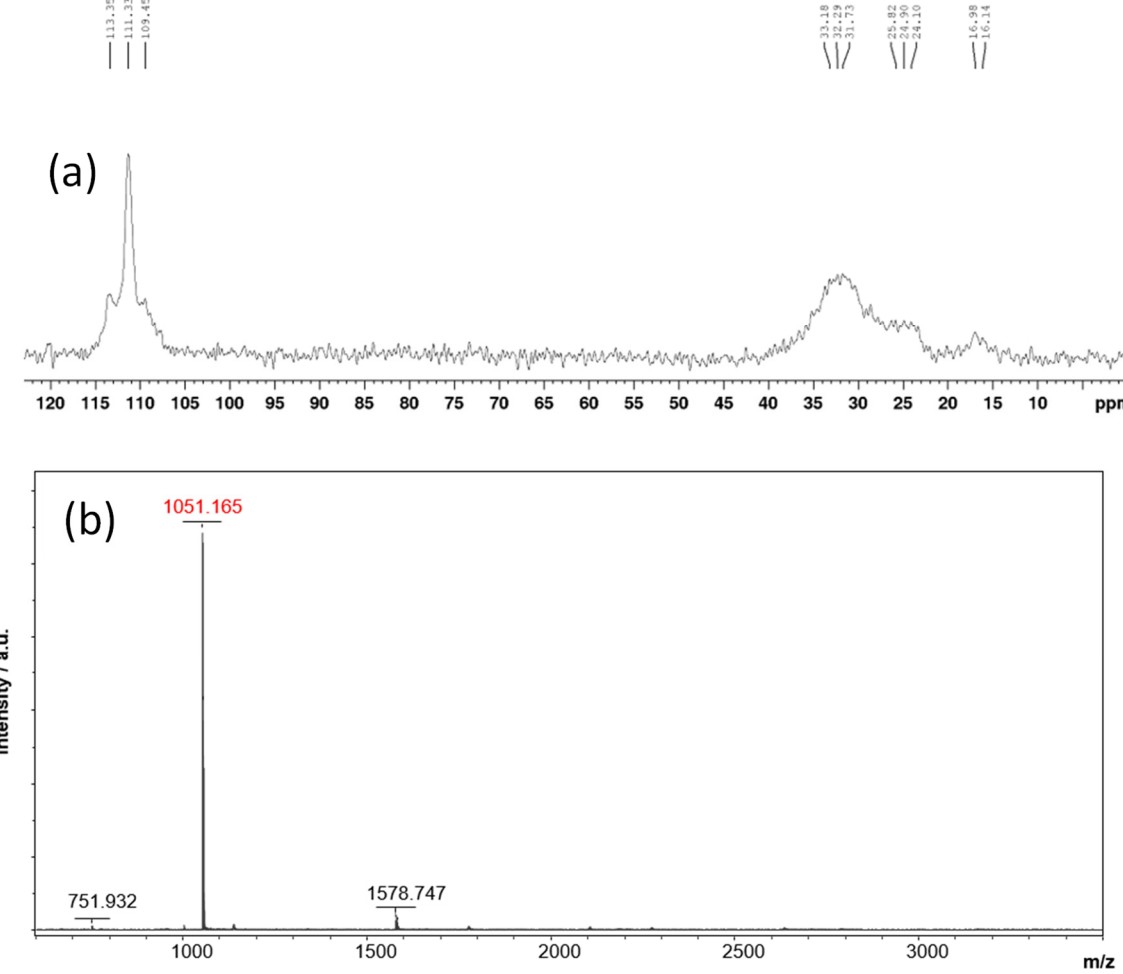

**Figure 1.** (**a**) Solid-state $^{13}$C NMR spectrum and (**b**) MALDI-TOF mass spectrum of [Ni]$_2$[L]$_2$ **2**.

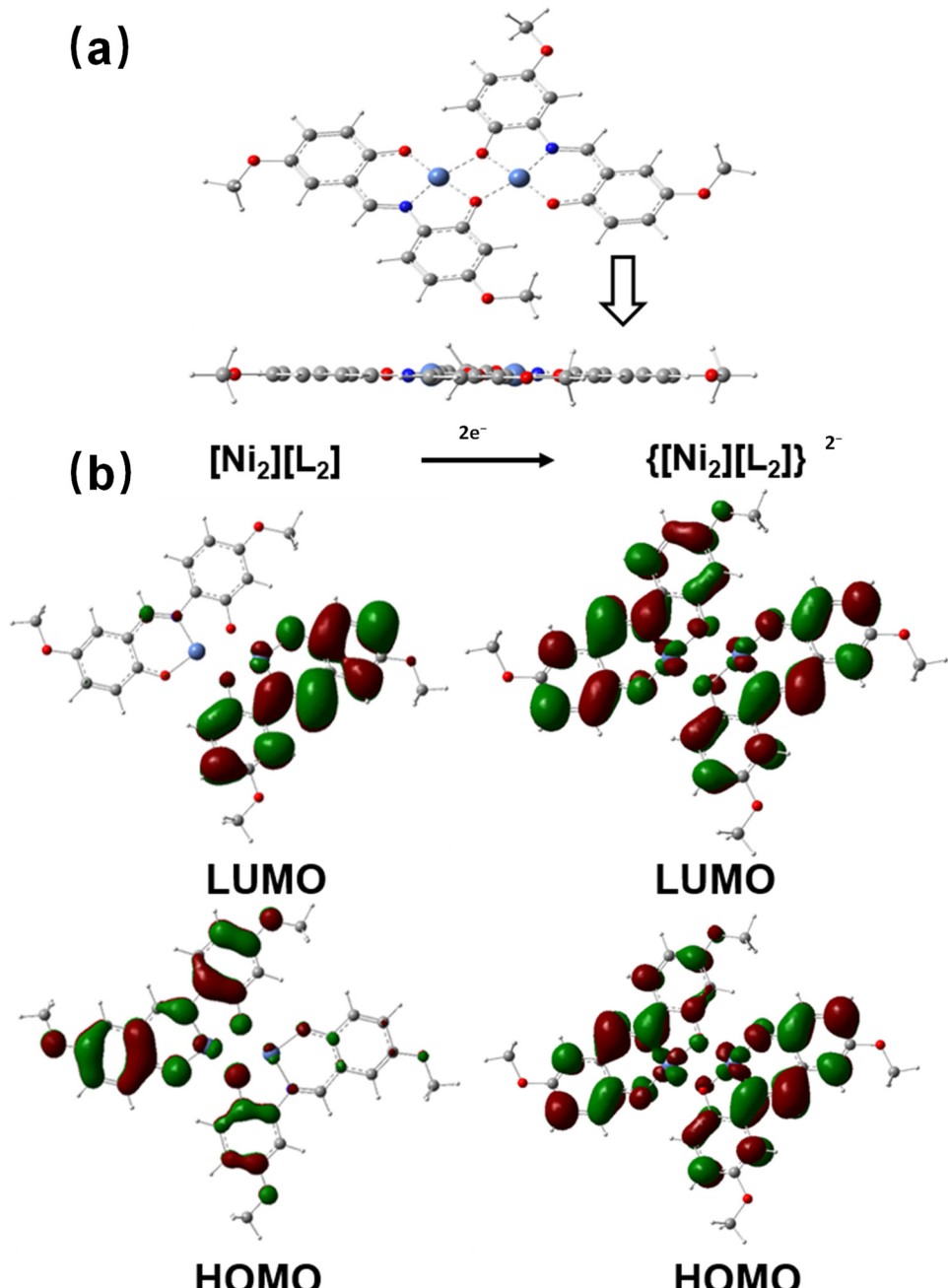

**Figure 2.** (**a**) Molecular structure (top and side view) of [Ni]$_2$[L]$_2$ **2**. (**b**) Frontier orbital maps of [Ni]$_2$[L]$_2$ **2** and reduced {[Ni]$_2$[L]$_2$ }$^{2-}$ complex obtained from DFT calculations. (Grey balls, carbon; white balls, hydrogen; red balls, oxygen; blue balls, nitrogen).

The drawback in the structural design of the catalyst is caused by issues of surface-layer morphology as aggregation hinders electrochemical performance [16]. To amend this, we employ the -O-C$_8$H$_{17}$ side chains to the ligand and catalyst structure, providing steric bulk and greater spatial conformation on the surface layer of electrodes. By employing computational modeling through Density Functional Theory (DFT) calculations, we have effectively visualized the optimized molecular geometry and the configurations of the Highest Occupied Molecular Orbital (HOMO) and Lowest Unoccupied Molecular Orbital (LUMO) for two important complexes: [Ni]$_2$[L]$_2$ **2** and its reduced counterpart, {[Ni]$_2$[L]$_2$}$^{2-}$ (Figure 2a,b). Our findings reveal that the catalyst adopts a planar geometry with two open axes available for the co-ordination of hydrogen-producing substrates.

Comparing these two complexes, we observe a significant reduction in the HOMO–LUMO energy gap for the electrochemically reduced $\{[Ni]_2[L]_2\}^{2-}$ species, which is now only 0.4 eV. The HOMO of this species is characterized by a substantial 40% contribution from Ni $d_{yz}$ orbitals, while there is an increased presence of $p_z$ orbitals at the two imine nitrogen atoms.

Our DFT calculations clearly indicate that this altered electronic environment, following a two-electron reduction, is highly favorable for the activation and binding of $H^+$, making it a promising candidate for subsequent hydrogen evolution reaction (HER) processes (Figure 2b).

XPS analyses detailing the elemental and electronic states of $[Ni]_2[L]_2$ **2** are displayed in Figure 3. The calculated atomic percentages obtained from the XPS spectrum are in good agreement with elemental analysis characterization for Ni, O, N, and C (Figure S13). Figure 3b provides narrow scans within the energy range of 850 eV to 885 eV, specifically focusing on the Ni2p3/2 and Ni2p1/2 peaks. In this range, we can attribute the peaks at 856 eV and 874 eV to the presence of Ni (II) in the $[Ni]_2[L]_2$ complex **2**. Additionally, the appearance of satellite peaks at 879 eV and 862 eV further supports the presence of Ni (II) [17]. These findings align with prior research that has characterized the electrochemical properties of molecular nickel-centered complexes [18]. For a more detailed examination of bond energies associated with various carbon–heteroatom linkages, please refer to Figure 3b. In this figure, three distinct peaks are observed: at 288.89 eV (corresponding to C-O bonds), 286.67 eV (representing C=N bonds), and another peak associated with carbon–carbon (-C=C- and -C-C-) bond linkages. These peaks can be confidently assigned to the relevant heteroatom and carbon bond linkages within $[Ni]_2[L]_2$ **2** [19].

The electrocatalytic activity of $[Ni]_2[L]_2$ was initially determined by conducting cyclic voltammetry (CV) (Figure 4a). In accordance with established electrochemical protocols, a standard three-electrode configuration was employed. Platinum (Pt) was judiciously chosen as the counter electrode (CE). As the centerpiece of our inquiry, the catalytically salient compound, $[Ni]_2[L]_2$, was systematically immobilized onto a carbon paper substrate, meticulously designated as the working electrode (WE). In this methodically orchestrated electrochemical investigation, the reference electrode of preference was the Ag/AgCl electrode, facilitating the establishment of a highly stable and well-defined electrochemical reference potential. The entirety of the electrochemical system was enveloped within an electrolyte solution, the composition of which was diligently prepared to consist of 0.5 M sulfuric acid ($H_2SO_4$) under an argon-saturated environment, thereby ensuring an oxygen-free milieu. The pH of this electrolytic medium was rigorously controlled to a value of 1.8, congruent with the precise requirements of the experimental design. Concurrently, the experimental setup was maintained at ambient room temperature. Figure 4a reveals a small anodic peak at −0.02 V vs. reversible hydrogen electrode (RHE) and an obvious anodic peak at a potential of −0.4 V vs. RHE, showing hydrogen evolution activity. The linear sweep voltammogram (LSV) presented in Figure 4b and Figure S11 further supports the excellent HER performance of $[Ni]_2[L]_2$.

To probe the stability of $[Ni]_2[L]_2$ **2**, a carbon paper spray coated with the electrocatalyst is characterized via chronoamperometric studies in 0.5 M $H_2SO_4$ purged with argon at room temperature. Figure 5a reveals long-term durability as catalytic activity is retained after 80 h of continued electrolysis.

Electrochemical impedance spectroscopy (EIS) served as the analytical tool of choice for the comprehensive examination and characterization of the electrochemical setup involving $[Ni]_2[L]_2$ **2**. The Bode plot, graphically depicting the dynamic response of the two-electrode system, has been elucidated in Figure 5b, and quantitative data have been compiled in Table 1. In the Pt–Pt two-electrode system, we quantified the resistance of the 0.5 M $H_2SO_4$ solution at pH 1.8, denoted as $R_{sol}$, revealing a measurement of 3.23 ohms. Furthermore, our scrutiny extended to the assessment of the Nafion membrane (NF) resistance, yielding $1.40 \times 10$ ohms. Within the constructed electrical circuit, these resistive elements, representing the electrolyte and the Nafion membrane, were interconnected in a

series arrangement. Expanding our analysis to the working electrodes (WE), inclusive of both the uncoated carbon paper (Cp) electrode and its counterpart coated with $[Ni]_2[L]_2$ **2**, we ascertained resistance values of $6.02 \times 10^4$ and $5.922 \times 10^4$ ohms, respectively. To account for the nonideal capacitive behavior inherent to the Cp electrode, we introduced a constant phase element (CPE) into the circuit model. Consequently, the resistance of $[Ni]_2[L]_2$ **2** was established at $1.257 \times 10^2$ ohms, and the corresponding capacitance of the $[Ni]_2[L]_2$ **2** coated Cp electrode was computed as $4.386 \times 10^{-6}$ F.

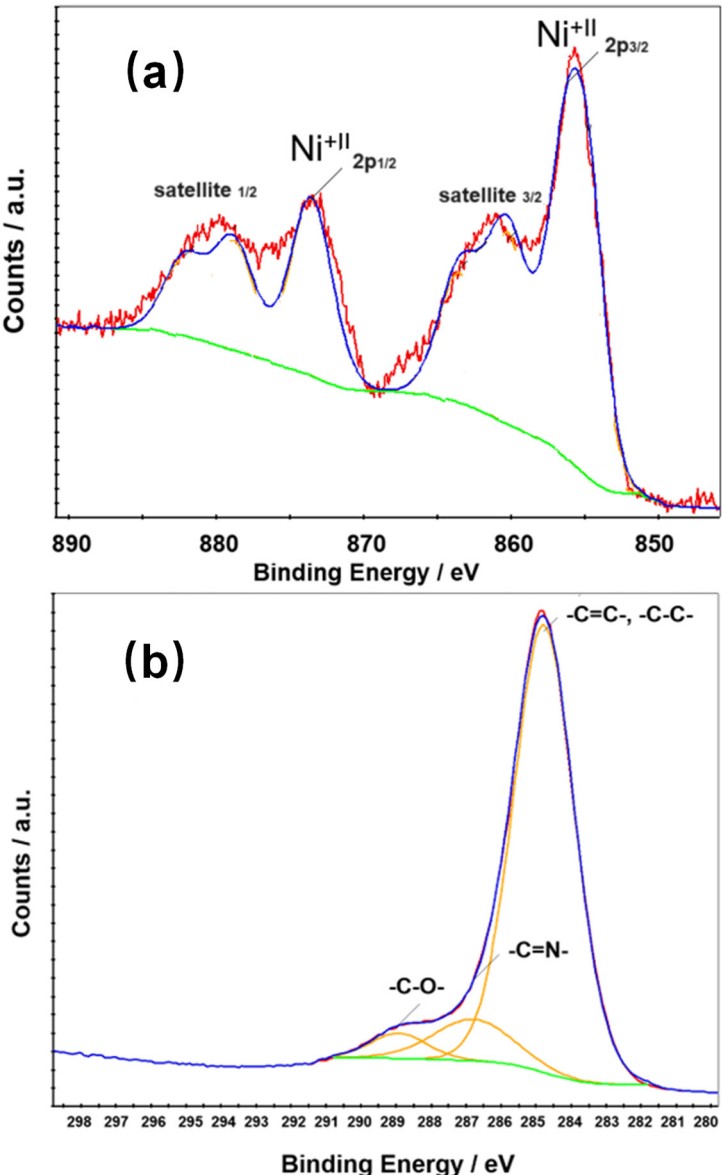

**Figure 3.** (**a**) XPS scans for Ni2p$_{3/2}$ and Ni2p$_{1/2}$ present in $[Ni]_2[L]_2$ **2**. (**b**) XPS scans for carbon (C1s) present in $[Ni]_2[L]_2$ **2**.

Figure S14 illustrates a Nyquist plot where the resistance values derived from Electrochemical Impedance Spectroscopy (EIS) manifest as distinctive features representing the specific circuit configuration [20,21]. Our analysis of the electrochemical impedance spectroscopy results enabled us to define the characteristics of the two-compartment setup, highlighting a significant finding of minimal system losses. These electrochemical circuit models, in conjunction with the insights provided by the Nyquist plot illustrating impedance behavior, serve as essential reference points for understanding the dynamics of electrochemical processes.

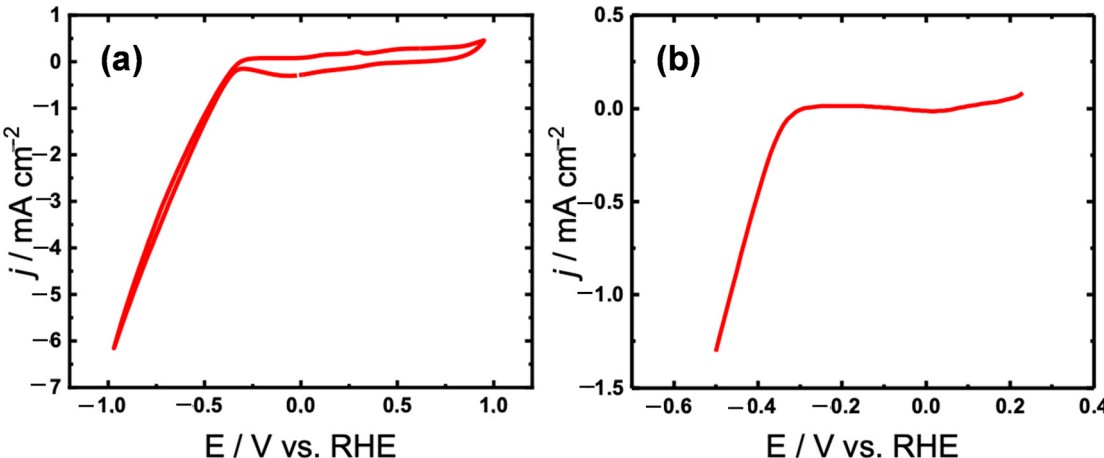

**Figure 4.** (**a**) Cyclic voltammogram (CV) and (**b**) linear sweep voltammogram (LSV) of $[Ni]_2[L]_2$ is loaded on carbon paper as working electrode (WE) and Ag/AgCl as a reference electrode in argon-saturated 0.5 M $H_2SO_4$ (pH 1.8) electrolyte at room temperature.

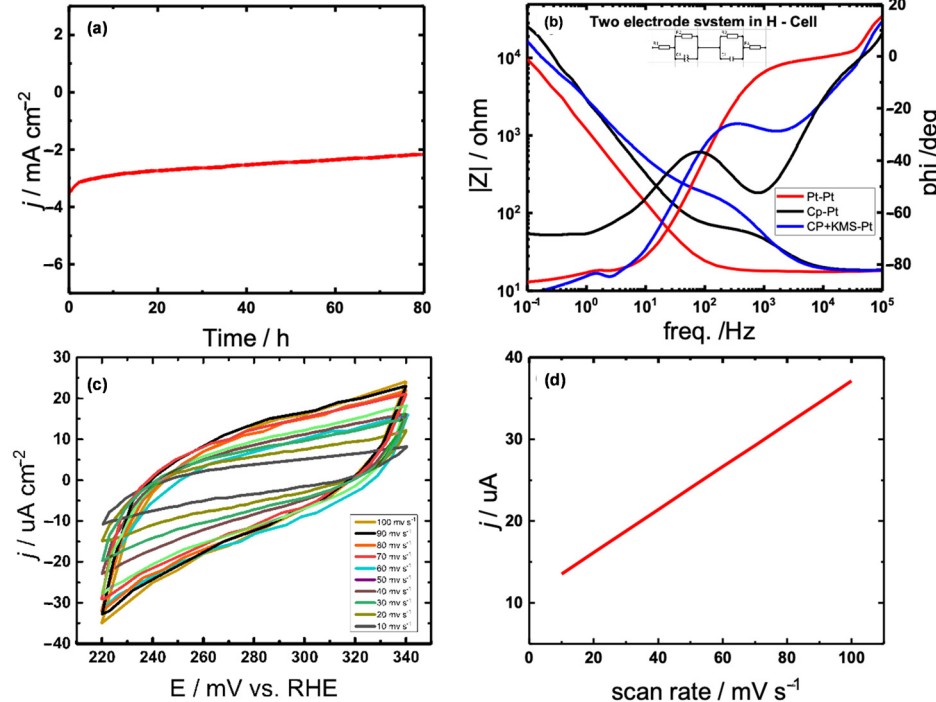

**Figure 5.** (**a**) Prolonged electrolysis for 80 h via chronoamperometry to determine long-term stability of $[Ni]_2[L]_2$ **2**. (**b**) Bode plot recorded via electrochemical impedance spectroscopy in the frequency range of $1 \times 10^{-1}$ Hz to $1 \times 10^5$ with a perturbation amplitude of 10 mV Hz. (**c**) Double-layer capacitance measurements for determining electrochemically active surface area for $[Ni]_2[L]_2$ **2** catalyst in 0.5 M $H_2SO_4$. (**d**) Linear dependence of the cathodic peak for current versus scan rate.

**Table 1.** Cell parameter extracted via electrochemical impedance spectroscopy measurements for HER.

| WE | CE | $R_{Sol}/\Omega$ | $R_{carrier}/\Omega$ | $R/[Ni]_2[L]\ 2\Omega$ | $R_{Me}/\Omega$ | C/F | CPE-T | CPE-P |
|---|---|---|---|---|---|---|---|---|
| Pt | Pt | 3.91 | $3.840 \times 10^4$ | - | $1.40 \times 10$ | - | $1.511 \times 10^{-4}$ | $9.433 \times 10^{-1}$ |
| Cp | Pt | 3.72 | $6.020 \times 10^4$ | - | $1.40 \times 10$ | - | $3.384 \times 10^{-5}$ | $9.645 \times 10^{-1}$ |
| $[Ni]_2[L]_2$ **2** | Pt | 3.83 | $5.922 \times 10^4$ | $1.257 \times 10^2$ | $1.40 \times 10$ | $4.386 \times 10^{-6}$ | $7.727 \times 10^{-5}$ | $8.151 \times 10^{-1}$ |

The electrochemically active surface area (ECSA) is calculated according to the electrochemical double-layer capacitance ($C_{dl}$), based on the correlation between the ECSA and $C_{dl}$ and $C_s$, where $C_s$ is the specific capacitance, by the equation ECSA = $C_{dl}/C_s$, resulting in 0.00002167 cm$^2$ [22]. The double-layer capacitance is determined using CV in 0.5 M $H_2SO_4$ in a range of potentials from 220 to 340 mV at varying scan rates beginning at 10 mV s$^{-1}$ to 100 mV s$^{-1}$ (Figure 5c). The value of $C_{dl}$ was determined by extracting the value from the slope of the current density plotted as a function of scan rate (Figure 5d). After characterization of [Ni]$_2$[L]$_2$ electrocatalytic capability, chronoamperometry was conducted for hour-long periods at varying potentials to assess the volume and efficiency of $H_2$ production (Table S1). From the results, we see that the catalyst achieved 95 $\pm$ 5% FE after 1 h of electrolysis at the overpotential of $-400$ mV. While electrolysis at both lower and higher potentials yielded $H_2$ production, $-400$ mV was determined to be the optimal working potential. From the volume of $H_2$ produced, the TOF and TON values were determined to be 0.214 and 0.0006 s$^{-1}$, respectively.

Table 2 presents various HER catalysts and their accompanying electrochemical conditions in comparison to [Ni]$_2$[L]$_2$ [9,23–25]. In comparison, [Ni]$_2$[L]$_2$ further establishes nickel as a viable alternative to platinum materials for efficient HER catalysis. Continued tuning of catalyst activity via ligand modification and electrochemical conditions will augment the application of [Ni]$_2$[L]$_2$ in large-scale $H_2$ production.

**Table 2.** Electrochemical conditions and activity of various hydrogen evolution reaction catalysts.

| Catalyst | Electrolyte | FE% | Overpotential V (vs. RHE) | Reference |
|---|---|---|---|---|
| PtCu Nanoparticles | $H_2SO_4$ | 95 $\pm$ 5 | $-0.01$ | Li et al. [23] |
| Ni(dcpdt)$_2$]$^{2-}$ | Acetate Buffer | 92 − 100 | $-0.37$ | Koshiba et al. [25] |
| [Ni(P$^{Ph}_2$N$^{Ph}$)$_2$] (BF$_4$)$_2$ | [(DMF)H]OTf + $H_2O$ | 95 $\pm$ 5 | $-0.63$ | Helm et al. [24] |
| [(Am1Py4)NiL](BF$_4$)$_2$ | $H_2O$ | 91 | $-1.25$ | Zhang et al. [9] |
| **[Ni]$_2$[Li]$_2$** | **$H_2SO_4$** | **95 $\pm$ 5** | **$-0.40$** | |

Mechanistic pathways for the hydrogen evolution reaction (HER) have been reported in the literature and, based on various reports, we propose several plausible mechanisms [26,27]. Initially, in the presence of the pristine catalyst with Ni (II) centers, the catalyst undergoes a reduction process, leading to the formation of Ni (I)-Ni (I) species (as illustrated in Figure 2b).

One proposed pathway involves the protonation of the imine-N, which brings a proton (H$^+$) into close proximity to the active Ni (I) center, thereby facilitating the reduction process. During this reduction step, a single electron transfer (SET) occurs from Ni (I) to the proton, resulting in the formation of a hydrogen radical intermediate. This hydrogen radical subsequently spontaneously recombines to produce a hydrogen molecule ($H_2$).

Another plausible mechanism is the oxidative protonation to form a hydride complex {Ni (III)-H}. This Ni (III)-H hydride complex either reacts with a proton to produce $H_2$ and Ni (III) complex (heterolytic route) or is further reduced to Ni (II)-H. This Ni (II)-H can react with a proton to give Ni (II) and $H_2$ (heterolytic route) or react with the neighboring Ni (II)-H to produce $H_2$ and the Ni (I)-Ni (I) complex (homolytic route). Another way is that Ni (III)-H underwent intramolecular homolytic reaction to produce $H_2$ and the original starting Ni (II)-Ni (II) complex.

It can be concluded that, once the hydride complex is formed, it can liberate hydrogen and, by simultaneous reduction, it returns to the initial catalyst to continue the cycle. This suggests that the key intermediate is Ni (III)-H complexes in a mechanistic cycle of HER.

## 3. Materials and Methods

### 3.1. Synthesis and Characterization

Chemicals were purchased from Fluka (Merck KGaA, Darmstadt, Germany), Alfa Aesar (Heysham, Britain), Sigma-Aldrich (St. Louis, MO, USA), and Merck. Commercially (Darmstadt, Germany) available hydroquinone was purchased from Sigma-Aldrich.

4-(octyloxy)phenol, 2-hydroxy-5-(octyloxy)benzaldehyde, 2-nitro-4-(octyloxy)phenol, 2-amino-4-(octyloxy)phenol, (Z)-2-((2-hydroxy-5-(octyloxy)benzylidene)amino)-4-(octyloxy)phenol, and [Ni]$_2$[L]$_2$ were synthesized according to reported literature procedures [28–31]. DCM was obtained using an M. Braun Inert Gas System GmBH (Bordeaux, France), where it is stored over a molecular sieve MB-SPS-7 (Merck KGaA, Darmstadt, Germany) under argon atmosphere. NMR solvents were purchased from Sigma-Aldrich. TLC was performed on Macherey-Nagel (Valencienner Str. 11, 52355 Düren, Germany) silica gel 60 (0.20 mm) with fluorescent indicator UV254 on aluminum plates and on Merck aluminum oxide 60 (0.20 mm) with fluorescent indicator UV254 on aluminum plates (Merck KGaA, Darmstadt, Germany). For chromatography, silica gel columns were prepared with silica gel 60 (0.070–0.20 mesh) from Grace (Kreuzauer Str. 46, 52355 Düren, Germany) and aluminum oxide 60, basic, activity level II from Fisher Scientific GmbH (Im Heiligen Feld 17, 58239 Schwerte, Germany). $^1$H and $^{13}$C NMR spectra were recorded on a Bruker (Karlsruhe, Germany) DRX 500 MHz spectrometer and a Bruker Advance 300 MHz NMR spectrometer. Chemical shifts are given in parts per million (ppm) on the delta scale ($\delta$) and are referenced to the used deuterated solvent for $^1$H-NMR. High-resolution mass spectra were obtained using an Agilent 6520 Q- TOF mass spectrometer (Santa Clara, CA, USA) with an ESI source and an Agilent G1607A coaxial sprayer and a Thermo Fisher Scientific LTQ Orbitrap XL (Waltham, MA, USA) with an Ion Max API Source. MALDI-TOF was measured on a Bruker Autoflex III smartbeam. UV-Vis absorption spectra were collected on a Varian CARY 300 Bio spectrophotometer (Agilent, Santa Clara, CA, USA) from 200 to 900 nm.

### 3.1.1. Synthesis of 4-(Octyloxy)phenol

Hydroquinone (4.0 g, 36.3 mmol) and potassium carbonate (5.0 g, 36.3 mmol) were added to a flame-dried round-bottom flask with a magnetic stir bar and dissolved in acetonitrile (100 mL) before being left to stir at room temperature for 30 min. After, 1-bromooctane (7.0 g, 36.3 mmol) was added dropwise and refluxed at 82 °C for 24 h. The reaction mixture was then left to cool to room temperature before being filtered through glass frit and washed with dichloromethane. The crude product was evaporated to dryness before purification via column chromatography (silica, EtOAc/heptane, [1:4]) to provide 4-(octyloxy)phenol (3.23 g, 40%) as a dark brown solid. R$_f$ = 0.5, $^1$H-NMR (300 MHz, CDCl3, 25 °C): $\delta$ = 6.82 (m, 4H, Ar-H), 4.68 (s, 1H, Ar-OH), 3.90 (t, $J$ = 6.6 Hz, 2H, -OCH$_2$-), 1.75 (m, 2H, -OCH$_2$<u>CH$_2$</u>-), 1.38-1.25 (m, 10H, -OCH$_2$CH$_2$<u>(CH$_2$)$_5$</u>), 0.90 (m, 3H, -OCH$_2$CH$_2$(CH$_2$)$_5$<u>CH$_3$</u>).

### 3.1.2. Synthesis of 2-Hydroxy-5-(octyloxy)benzaldehyde

Magnesium chloride (1.7 g 17.9 mmol) and paraformaldehyde (0.9 g, 30.0 mmol) were added to a flame-dried round-bottom flask with a magnetic stir bar and dissolved in THF (100 mL). Triethylamine (1.8 g, 17.8 mmol) was then added, and the solution was left to stir at room temperature for 10 min. After, 4-(octyloxy)phenol (1.1 g, 5 mmol) was added and refluxed at 66 °C for 24 h. The reaction mixture was then left to cool before being diluted with ether and washed with 1 M HCl. The mixture was dried with magnesium sulfate and evaporated under reduced pressure to afford 2-hydroxy-5-(octyloxy)benzaldehyde (1.1 g, 88%) as a brown oil. $^1$H-NMR (300 MHz, CDCl3, 25 °C): $\delta$ = 10.62 (s, 1H, -OH), 9.82 (s, $^1$H, -CHO), 7.12 (dd, $J$ = 9.0, 3.1 Hz, 1H, Ar-H), 6.98 (d, $J$ = 3.1 Hz, 1H, Ar-H), 6.90 (d, $J$ = 9.1 Hz, 1H, Ar-H), 3.92 (t, $J$ = 6.6 Hz, 2H, -OCH$_2$-), 3.90 (t, 2H, -OCH$_2$-), 1.75 (m, 2H, -OCH$_2$<u>CH$_2$</u>-), 1.38-1.25 (m, 10H, -OCH$_2$CH$_2$<u>(CH$_2$)$_5$</u>), 0.90 (m, 3H, -OCH$_2$CH$_2$(CH$_2$)$_5$<u>CH$_3$</u>).

### 3.1.3. Synthesis of 2-Nitro-4-(octyloxy)phenol

4-(octyloxy)phenol (3.0 g, 13.0 mmol) was added to benzene (100 mL) in a flame-dried round-bottom flask with a magnetic stir bar. Once cooled to 0 °C, 56% aqueous nitric acid solution (13 mmol) was added slowly and vigorously stirred for 2 min. The reaction mixture was then quenched with water, after which the organic phase was dried with magnesium sulfate and evaporated to reveal the crude product as an orange powder.

Impurities were removed via column chromatography (silica, EtOAc/heptane, [1:20]) to provide 2-nitro-4-(octyloxy)phenol (1.94 g, 56%) as an orange solid. $R_f$ = 0.4, [1]H-NMR (300 MHz, CDCl3, 25 °C): δ = 10.33 (s, 1H, -OH), 7.50 (d, *J* = 3.07, 1H, Ar-H), 7.22 (dd, $J_1$ = 3.13, $J_2$ = 3.04, 1H, Ar-H), 7.08 (d, *J* = 9.17, 1H, Ar-H), 3.94 (t, 2H, -OCH$_2$-), 1.75 (m, 2H, -OCH$_2$CH$_2$-), 1.50-1.25 (m, 10H, -OCH$_2$CH$_2$(CH$_2$)$_5$), 0.90 (m, 3H, -OCH$_2$CH$_2$(CH$_2$)$_5$CH$_3$).

### 3.1.4. Synthesis of 2-Amino-4-(octyloxy)phenol

In a flame-dried round-bottom flask equipped with a stir bar, a solution of 2-nitro-4-(octyloxy)phenol (1.3 g, 4.8 mmol) and 10% Pd/C (51 mg) in MeOH (50 mL) was placed under hydrogen atmosphere. The reaction mixture was left to stir for 24 h. After, the catalyst was filtered off and evaporated to reveal 2-amino-4-(octyloxy)phenol (0.99 g, 89%) as a dark red solid. ([1]H-NMR (300 MHz, CDCl3, 25 °C): δ = 6.65 (d, *J* = 8.47, 1H, Ar-H), 6.35 (d, *J* = 2.77, 1H, Ar-H), 6.21 (dd, $J_1$ = 2.79, $J_2$ = 2.78, 1H, Ar-H), 3.87 (m, 2H, -NH$_2$), 3.94 (t, 2H, -OCH$_2$-), 1.75 (m, 2H, -OCH$_2$CH$_2$), 1.50-1.25 (m, 10H, -OCH$_2$CH$_2$(CH$_2$)$_5$), 0.90 (m, 3H, -OCH$_2$CH$_2$(CH$_2$)$_5$CH$_3$).

### 3.1.5. Synthesis of (Z)-2-((2-Hydroxy-5-(octyloxy)-benzylidene)amino)-4-(octyloxy)phenol **1**

2-amino-4-(octyloxy)phenol (0.4 g, 1.68 mmol) was dissolved in ethanol (20 mL) in a flame-dried round-bottom flask equipped with a magnetic stir bar. Then, 2-hydroxy-5-(octyloxy)benzaldehyde (0.42 g, 1.68 mmol) was added dropwise to the reaction mixture and refluxed for 1 h at 80 °C. After, the reaction was left to cool and evaporated to dryness before purification via column chromatography (silica, EtOAc/heptane, [1:5]) (Z)-2-((2-hydroxy-5-(octyloxy)benzylidene)amino)-4-(octyloxy)phenol (0.55 g, 69%) as a black solid. ([1]H-NMR s(300 MHz, CDCl$_3$, 25 °C): δ = 8.61 (s, 1H, -CH=N-), 7.03 (dd, $J_1$ = 9.0, $J_2$ = 2.8 Hz, 1H, Ar-H), 7.00 (d, *J* = 7.5 Hz, 1H, Ar-H), 6.90 (m, 2H, Ar-H), 6.79 (dd, *J* = 8.8, $J_2$ = 2.8 Hz, 1H, Ar-H), 6.72 (d, *J* = 2.8 Hz, 1H, Ar-H), 3.92 (m, 4H, -OCH$_2$ -), 1.75 (m, 4H, -OCH$_2$CH$_2$ -), 1.50-1.25 (m, 20H, -OCH$_2$CH$_2$(CH$_2$)$_5$), 0.90 (m, 6H, -OCH$_2$CH$_2$(CH$_2$)$_5$CH$_3$). $\lambda_{max}$ nm: 386. FTIR (cm$^{-1}$): 2985, 2940, 2895, 2830, 1622, 1492, 1454, 1299, 1241, 1126, 1036, 808, 735, 563, 464.

### 3.1.6. Synthesis of [Ni]$_2$[L]$_2$ Complex **2**

(Z)-2-((2-hydroxy-5-(octyloxy)benzylidene)amino)-4-(octyloxy)phenol (0.1 g, 2 mmol) and triethyl amine (1.5 equiv.) were dissolved in ethanol (20 mL) in a flame-dried round-bottom flask equipped with a magnetic stir bar. After, a solution of nickel (II) acetate hexahydrate (50 mg, 0.2 mmol) in ethanol (20 mL) was added dropwise to the round-bottom flask and refluxed for 1 h at 65 °C. The reaction mixture was cooled before filtering to obtain the [Ni]$_2$[L]$_2$ as a dark red precipitate. The crude, solid complex was washed with hot ethanol twice to remove any impurities, revealing the pure [Ni]$_2$[L]$_2$ complex (70 mg, 33%). MALDI-TOF *m/z*: calcd. For $C_{58}H_{82}N_2Ni_2O_8^+$ 1051.478; found [M + H]$^+$: 1051.165. FTIR (cm$^{-1}$): 2920, 2852, 1600, 1545, 1489, 1468, 1262, 1202, 1151, 1046, 809, 512. Elemental Analysis: Anal. Calcd: C: 66.18, H: 7.85, N: 2.66%. Found: C: 65.26, H: 7.27, N: 2.55%.

**DFT Calculations.** All calculations were performed using the Gaussian 09 package version EM64L-G09RevC.01. Electronic structure calculations were based on Kohn–Sham density functional theory (KS-DFT) with Becke's three-parameter hybrid functional (B3LYP) and a compound basis set, where the Pople's 6–311 + G (d,2p) basis sets were used for C, H, N, O, and Ni. For our system, we first performed a tight structural optimization, followed by a frequency calculation to confirm that the optimized structure was indeed a minimum (with no imaginary frequencies).

**XPS measurements.** XPS was performed by using a Theta Probe, Thermo Fisher Scientific, Bishop Meadow Road, Loughborough, Leicestershire, UK, using monochromatic Al K$\alpha$ X-rays (hν = 1486.6 eV), spot size of 400 microns, and with a photoelectron take-off angle of 90° with respect to the surface plane. The binding energies were corrected using the C1s peak at BE = 284.6 eV that arises from adventitious hydrocarbon.

## 4. Conclusions

To conclude, we report on a highly efficient and unique binuclear nickel complex $[Ni]_2[L]_2$ for HER electrolysis. Metalation of a Schiff base ligand as a precursor allows for a facile and highly tunable synthetic pathway. The design of the electrocatalyst involves electronegative heteroatoms in direct adjacency to the nickel centers to garner a high affinity to protons in aqueous electrolyte. Characterization of the electrocatalytic ability of the catalyst revealed promising results, as a potential of −400 mV vs. RHE is comparable to highly efficient platinum-containing materials for HER. When applied in HER electrolysis via chronoamperometric measurements, the catalyst performed with high efficiency, providing $H_2$ product with near 100% FE at the potential of −400 mV vs. RHE. Prolonged electrolysis provided evidence for the long-term stability of the catalyst as well. As the standard for highly efficient HER catalyst for industrial use relies on rare earth platinum, we provide an environmentally friendly alternative with comparable efficiency and performance for HER electrolysis.

**Supplementary Materials:** The following supporting information can be downloaded at: https://www.mdpi.com/article/10.3390/catal13101348/s1, including characterization and spectral data of all intermediate compounds and $Ni_2L_2$ **2** ($^1$H NMR, $^{13}$C NMR, solid state $^{13}$C NMR, UV-vis, FT-IR), all heterogeneous electrochemical data acquired for $Ni_2L_2$ **2**. Figure S1. $^1$H-NMR spectrum of 4-(octyloxy)phenol in CDCl$_3$. Figure S2. $^1$H-NMR spectrum of 2-hydroxy-5-(octyloxy)benzaldehyde in CDCl$_3$. Figure S3. $^1$H-NMR spectrum of 2-nitro-4-(octyloxy)phenol in CDCl$_3$. Figure S4. $^1$H-NMR spectrum of 2-amino-4-(octyloxy)phenol in CDCl$_3$. Figure S5. $^1$H-NMR spectrum of (Z)-2-((2-hydroxy-5-(octyloxy)benzylidene)amino)-4-(octyloxy)phenol in CDCl$_3$. Figure S6. $^{13}$C NMR spectrum of $[Ni]_2[L]_2$ **2**. Figure S7. UV-VIS spectrum of (Z)-2-((2-hydroxy-5-(octyloxy)benzylidene)amino)-4-(octyloxy)phenol 1 in CH$_2$Cl$_2$. Figure S8. FTIR spectrum of (Z)-2-((2-hydroxy-5-(octyloxy)benzylidene)amino)-4-(octyloxy)phenol 1. Figure S9. FTIR spectrum of $[Ni]_2[L]_2$ **2**. Figure S10. MALDI-TOF Spectrum of $[Ni]_2[L]_2$ **2**. Figure S11. Chronoamperometric measurement of $[Ni]_2[L]_2$ **2** at −400 mV vs. RHE for 1 h. Figure S12. Chronoamperometric measurement of carbon black on carbon paper at −400 mV vs. RHE for 1 h. Figure S13. LSV measurement of $[Ni]_2[L]_2$ **2** from 0.2 to −0.7 V vs. RHE. Figure S14. Nyquist plot of $[Ni]_2[L]_2$ **2**. Table S1: Cell parameter extracted via electrochemical impedance measurements for HER. Table S2: Faradaic efficiencies of $H_2$ production after chronoamperometry using $[Ni]_2[L]_2$ **2** as WE. Table S3. The calculated atomic percentages obtained from XPS spectra for Ni, O, N, and C [14,19,28–31].

**Author Contributions:** Conceptualization: W.S.; methodology: K.S., H.A., A.A., F.Y. and W.S.; validation: W.S.; formal analysis: K.S., H.A., A.A., F.Y. and W.S.; investigation: K.S., H.A., F.Y., A.A. and W.S.; writing—original draft preparation: K.S., A.A. and W.S.; writing—review and editing: K.S. and W.S.; visualization: K.S., A.A. and W.S.; supervision: W.S.; funding acquisition: K.S. and W.S. All authors have read and agreed to the published version of the manuscript.

**Funding:** W.S. acknowledges the financial support of the Austrian Science Fund (FWF Standalone Projects P28167 "Heterogeneous catalysis for water oxidation and hydrogen evolution" and P32045 "Catalysts for biomass valorization") and of the Austrian Research Promotion Agency FFG (FFG Project Nr.: 883671). The NMR spectrometers were acquired in collaboration with the University of South Bohemia (CZ) with financial support from the European Union through the EFRE INTERREG IV ETC-AT-CZ program (project M00146, "RERI-uasb"). K.S. acknowledges the financial support of the Fulbright-Austrian Marshall Plan Foundation Award in Science and Technology. Open Access Funding by the Austrian Science Fund (FWF P32045 "Catalysts for biomass valorization").

**Data Availability Statement:** Not applicable.

**Acknowledgments:** We gratefully thank Markus Himmelsbach from the Department of Analytical Chemistry and Clemens Schwarzinger from the Institute for Chemical Technology of Organic Materials at the JKU for carrying out ESI HR-MS and MALDI-TOF MS measurements.

**Conflicts of Interest:** The authors declare no conflict of interest.

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
