# Peer review of "A Molecular Binuclear Nickel (II) Schiff Base Complex for Efficient HER Electrocatalysis"

_catalysts, doi:10.3390/catal13101348_

Round 1

Reviewer 1 Report

In this manuscript, the authors reported the synthesis of a molecular binuclear nickel (II/III) schiff base complex and its application for hydrogen evolution reaction. The catalyst displays excellent HER activity with 100% faradaic efficiency (FE) at an onset potential of 0.4 V vs. reverse hydrogen electrode (RHE), sustained catalytic activity during prolonged electrolysis for greater than 17 hours, and provided a turn over number (TON) of 0.0006 s-1. This work can be accepted by the journal after major revision.

1. The English in the manuscript should be highly refined.

2. The long-term durability test of catalyst only lasts for 17.5 h, and a longer stability test should be provided.  

3. In this work, the hydrogen evolution performance of the same type of catalysts should be listed and compared.

4. The word and pictures inset the Figure 3b and Figure 5b should be clearer.

The English in the manuscript should be highly refined.

Author Response

Comments to Referee 1:

1. The English in the manuscript should be highly refined.

Thank you for your feedback regarding the quality of the English in our manuscript. We appreciate your attention to detail and your commitment to maintaining high standards of language proficiency in scientific publications. We have taken the necessary steps to address this concern. Specifically, we have thoroughly revised the manuscript, focusing on refining the language and improving its overall readability. We believe that these revisions will significantly enhance the clarity and precision of our manuscript, ultimately improving its accessibility.

2. The long-term durability test of catalyst only lasts for 17.5 h, and a longer stability test should be provided.  

Thank you for your comment regarding the long-term experiment. We have repeated the experiment now for 80 hrs. and added the graph into the manuscript.

3. In this work, the hydrogen evolution performance of the same type of catalysts should be listed and compared.

Thank you for your valuable feedback and suggestion regarding the comparison of the hydrogen evolution performance of the same type of catalysts in our manuscript. We appreciate your insight into enhancing the comprehensiveness of our work. We agree that providing a comparative analysis of the hydrogen evolution performance of the same type of catalysts is a crucial aspect of electrocatalysis research. In our revised manuscript, we incorporated a table which highlights the comparative evaluation of the catalysts used in our study with various Nickel containing molecular catalysts. In addition, the performance of state-of-the-art platinum nanoparticles is included within this table to address any lack of information regarding the performance of platinum in HER catalysis.

4. The word and pictures inset the Figure 3b and Figure 5b should be clearer.

Thank you for your feedback regarding the clarity of the word and pictures inset in Figure 3b and Figure 5b in our manuscript. We appreciate your attention to the visual presentation of our data. In response to your comment, we have made significant improvements to the clarity of these figures. Furthermore, we have expanded the section in our manuscript that discusses the use of electrochemical impedance spectroscopy (EIS) as a methodology. This includes a more comprehensive explanation of the principles and procedures involved in conducting EIS experiments, as well as the specific conditions and parameters used in our study. We believe that this additional information will provide readers with a clearer understanding of the methodology and its application in our research. We hope that these revisions and additions will significantly enhance the clarity and accessibility of our figures and the associated explanations.

Reviewer 2 Report

Report on the manuscript catalysts-2611559 entitled “A Molecular Binuclear Nickel (II/III) Schiff Base complex for Efficient HER Electrocatalysis”.

The submitted manuscript should be revised. This work represented the hydrogen evolution reaction by a binuclear nickel (II/III) Schiff base complex through the presence of electronegative heteroatoms in proximity to the metal centers augments proton affinity. The introduced catalyst has HER activity with 100% faradaic efficiency (FE) at an onset potential of 0.4 V vs. reverse hydrogen electrode. In short, the following points should be addressed:

1.   The language of the manuscript should be revised.

2.   In the structure of Schiff base, Ni ions were neutralized via OH phenolic group and so, become only O-Ni but the XPS and the suggested structure has II and III Ni. Please, explain this in the manuscript?

3.   “Furthermore, orbital depictions show ideal configurations for substrate binding thus giving molecular insight to the enhanced HER activity of the catalyst design.”, this statement should have reference and more explanation.

4.   “The spectra are in agreement with elemental analysis characterization as the presence of Ni, O, N, and C are present.”, the XPS is surface characterization and elemental analysis is bulk analysis so, it should have difference, please, explain in the manuscript.

5.   Table S1 including the impedance parameters is very important and so, it should be in the main manuscript. Please, revise calculations because R sol is the same for Pt and Schiff base. It’s too strange that Pt has the same resistance of Schiff base!

6.   The impedance analysis should have Nyquist plots beside the Bode phase or total impedance and the equivalent circuit should be explained in detail with the reason of chosen this equivalent circuit [Suggested references for this: Journal of Electroanalytical Chemistry, Volume 785, 2017, Pages 190-195 & Journal of Alloys and Compounds, Volume 816, 2020, 152513].

7.   The details of electrochemical part should be clear in the experimental part of the submitted manuscript.

It should be revised.

Author Response

Comments to Referee 2:

  1.   The language of the manuscript should be revised.

Thank you for your feedback regarding the quality of the English in our manuscript. We appreciate your attention to detail and your commitment to maintaining high standards of language proficiency in scientific publications. We have taken the necessary steps to address this concern. Specifically, we have thoroughly revised the manuscript, focusing on refining the language and improving its overall readability. We believe that these revisions will significantly enhance the clarity and precision of our manuscript, ultimately improving its accessibility.

  1.   In the structure of Schiff base, Ni ions were neutralized via OH phenolic group and so, become only O-Ni but the XPS and the suggested structure has II and III Ni. Please, explain this in the manuscript?

After thorough inspection of reference XPS spectra, we came to the conclusion that the complex consists solely of Ni+II ions. We changed the manuscript accordingly.

  1.   “Furthermore, orbital depictions show ideal configurations for substrate binding thus giving molecular insight to the enhanced HER activity of the catalyst design.”, this statement should have reference and more explanation.

      We have added a new Figure 2b and added an extensive explanation to the manuscript:

The drawback in the structural design of the catalyst is caused by issues of surface layer morphology as aggregation hinders electrochemical performance[16]. To amend this, we employ the –O-C8H17 side-chains to the ligand and catalyst structure providing steric bulk and greater spatial conformation on the surface layer of electrodes. Through computational modeling via DFT calculations, the optimized geometry as well as the HOMO and LUMO orbital configurations of the [Ni]2[L]2 2 and the reduced {[Ni]2[L]2}2- complexes are visualized (Figure 2a,b). The model displays a planar geometry of the catalyst with two open axes for coordination of hydrogen producing substrates. Compared to complex [Ni]2[L]2 2, the electrochemically reduced {[Ni]2[L]2}2- species exhibits increased electron density near and at the metal centers which results in an electronic environment favorable for H+ activation/binding and subsequent HER.

Mechanistic pathways are reportedly known in the literature, based on different reports we propose the plausible mechanism. The pristine catalyst with the Ni(II, III) centers are reduced first to Ni(I)-Ni(I) species (Figure 2b), which then undergoes oxidative protonation to form a hydride complex {Ni(III)–H}. Spin delocalization onto the ligand framework likely stabilizes the NiIII−H species. This Ni(III)–H hydride complex either reacts with a proton to produce H2 and Ni(III) complex (heterolytic route), or is further reduced to Ni(II)–H. This Ni(II)–H can react with a proton to give Ni(II) and H2 ((heterolytic route), or react with the neighboring Ni(II)–H to produce H2 and the Ni(I)-Ni(I) complex (homolytic route). Another way is that Ni(III)–H underwent intramolecular homolytic reaction to produce H2 and the original starting Ni(II)-Ni(II) complex. It can be concluded that once the hydride complex is formed, it can liberate hydrogen, and by simultaneous reduction, it gets back to the initial catalyst to continue the cycle. This suggests that the key intermediate is Ni(III)–H complexes in a mechanistic cycle of HER.

  1.   “The spectra are in agreement with elemental analysis characterization as the presence of Ni, O, N, and C are present.”, the XPS is surface characterization and elemental analysis is bulk analysis so, it should have difference, please, explain in the manuscript.

Atomic percentages can be calculated quite accurately from XPS spectra and can then be compared to the bulk elemental analysis.

  1.   Table S1 including the impedance parameters is very important and so, it should be in the main manuscript. Please, revise calculations because R sol is the same for Pt and Schiff base. It’s too strange that Pt has the same resistance of Schiff base!

Thank you for your valuable feedback regarding the impedance parameters and calculations. The calculations for the resistance of the platinum and Schiff base catalyst have been revised and added to the manuscript using more sensitive measurements.

  1.   The impedance analysis should have Nyquist plots beside the Bode phase or total impedance and the equivalent circuit should be explained in detail with the reason of chosen this equivalent circuit [Suggested references for this: Journal of Electroanalytical Chemistry, Volume 785, 2017, Pages 190-195 & Journal of Alloys and Compounds, Volume 816, 2020, 152513].

Thank you for your valuable feedback and recommendations regarding the impedance analysis in our manuscript. We appreciate your attention to the details of our methodology and analysis. In response to your suggestion, we included Nyquist plot in the supporting information as well as a diagram of the circuit configuration used in our experimentation. We understand the importance of elucidating the rationale behind the selection of the equivalent circuit and its relevance to our electrochemical impedance spectroscopy (EIS) analysis. As a result, the added information is referenced in the text of the manuscript. Inferences made regarding the added supplementary figures are supported by the suggested references.

  1.   The details of electrochemical part should be clear in the experimental part of the submitted manuscript.

Thank you for your feedback regarding the clarity of the electrochemical part in the experimental section of our manuscript. We appreciate your focus on the transparency of our methodology. In response to your comment, we took steps to enhance the clarity of the electrochemical part in the experimental section. We recognized the importance of providing a detailed and clear description of our experimental procedures to ensure that other researchers could replicate our work accurately.

Reviewer 3 Report

The research topic is relevant, the data obtained look interesting. However, the manuscript contains a number of fundamental errors. A major revision is needed before this manuscript being accepted for publication. 

The main drawbacks are listed below.

1)    Figure 2b shows the calculated HOMO and LUMO orbital configurations. However, the text does not provide a detailed description of the structure of these orbitals and how they relate to the catalytic activity of the complex. I recommend to remove the pictures to the SI.

2)    Figure 3a shows the XPS survey spectrum, which does not provide any useful information. Moreover, the figure contains strange inscriptions like “Ni2p3” and “Ni2p4”, the meaning of which is not clear. I recommend to remove the picture to the SI as well.

3)    The description of the XPS data is very poorly written and cannot be published in its presented form. The authors do not understand the basics of the XPS method. I recommend hiring an expert in the field of XPS to write the article. For example, the authors write “the narrow scans of Ni2p1/2 and Ni2p3/2 in the 850 eV to 875 eV range”. This is a mistake. The Ni2p3/2 binding energy is always less than the Ni2p3/2 binding energy. The next sentence is also wrong. The authors write “the 856 eV peak can be assigned as NiIII to one of the nuclear nickel centers while the 874 eV peak can be assigned as NiII to the other accompanying nickel center”. In fact, the Ni2p spectrum is a Ni2p3/2 - Ni2p1/2 spin-orbital doublet and both of these peaks can be attributed to nickel in the Ni2+ state. Designations NiII and NiIII are incorrect; Ni2+ and Ni3+ must be used. Scheme I, Fig. 2 and the text should be corrected. The curve fit analysis of the Ni2p spectrum is uncorrected. The spectrum cannot be approximated by 7 peaks because the Ni2p spectrum must be approximated by several doublets, i.e. contain an even number of peaks.

4)    XPS data are very important for proving the structure of the synthesized complex, therefore all statements about the correlation of peaks must be confirmed by literature references.

Ыome terms are incorrect. The authors should check the text carefully. For example, "bond energy" should be replaced by "the binding energy".

Author Response

Referee 3:

  1. Figure 2b shows the calculated HOMO and LUMO orbital configurations. However, the text does not provide a detailed description of the structure of these orbitals and how they relate to the catalytic activity of the complex. I recommend to remove the pictures to the SI.

We have added a new Figure 2b and added an extensive explanation to the manuscript:

The drawback in the structural design of the catalyst is caused by issues of surface layer morphology as aggregation hinders electrochemical performance[16]. To amend this, we employ the –O-C8H17 side-chains to the ligand and catalyst structure providing steric bulk and greater spatial conformation on the surface layer of electrodes. Through computational modeling via DFT calculations, the optimized geometry as well as the HOMO and LUMO orbital configurations of the [Ni]2[L]2 2 and the reduced {[Ni]2[L]2}2- complexes are visualized (Figure 2a,b). The model displays a planar geometry of the catalyst with two open axes for coordination of hydrogen producing substrates. Compared to complex [Ni]2[L]2 2, the electrochemically reduced {[Ni]2[L]2}2- species exhibits increased electron density near and at the metal centers which results in an electronic environment favorable for H+ activation/binding and subsequent HER.

Mechanistic pathways are reportedly known in the literature, based on different reports we propose the plausible mechanism. The pristine catalyst with the Ni(II, III) centers are reduced first to Ni(I)-Ni(I) species (Figure 2b), which then undergoes oxidative protonation to form a hydride complex {Ni(III)–H}. Spin delocalization onto the ligand framework likely stabilizes the NiIII−H species. This Ni(III)–H hydride complex either reacts with a proton to produce H2 and Ni(III) complex (heterolytic route), or is further reduced to Ni(II)–H. This Ni(II)–H can react with a proton to give Ni(II) and H2 ((heterolytic route), or react with the neighboring Ni(II)–H to produce H2 and the Ni(I)-Ni(I) complex (homolytic route). Another way is that Ni(III)–H underwent intramolecular homolytic reaction to produce H2 and the original starting Ni(II)-Ni(II) complex. It can be concluded that once the hydride complex is formed, it can liberate hydrogen, and by simultaneous reduction, it gets back to the initial catalyst to continue the cycle. This suggests that the key intermediate is Ni(III)–H complexes in a mechanistic cycle of HER.

Figure 3a shows the XPS survey spectrum, which does not provide any useful information. Moreover, the figure contains strange inscriptions like “Ni2p3” and “Ni2p4”, the meaning of which is not clear. I recommend to remove the picture to the SI as well.

The description of the XPS data is very poorly written and cannot be published in its presented form. The authors do not understand the basics of the XPS method. I recommend hiring an expert in the field of XPS to write the article. For example, the authors write “the narrow scans of Ni2p1/2 and Ni2p3/2 in the 850 eV to 875 eV range”. This is a mistake. The Ni2p3/2 binding energy is always less than the Ni2p3/2 binding energy. The next sentence is also wrong. The authors write “the 856 eV peak can be assigned as NiIII to one of the nuclear nickel centers while the 874 eV peak can be assigned as NiII to the other accompanying nickel center”. In fact, the Ni2p spectrum is a Ni2p3/2 - Ni2p1/2 spin-orbital doublet and both of these peaks can be attributed to nickel in the Ni2+ state. Designations NiII and NiIII are incorrect; Ni2+ and Ni3+ must be used. Scheme I, Fig. 2 and the text should be corrected. The curve fit analysis of the Ni2p spectrum is uncorrected. The spectrum cannot be approximated by 7 peaks because the Ni2p spectrum must be approximated by several doublets, i.e. contain an even number of peaks.

Thank you for your valuable comments!

We have discussed the data with our in-house XPS specialists and obtained reference spectra for Ni+III and Ni+II complexes. We came to the conclusion that the [Ni2L2] 2 complex consists solely of Ni+II ions.

We have rewritten the paragraph of the manuscript.

Figure 3b shows the narrow scans of Ni2p3/2 and Ni2p1/2 in the 850 eV to 875 eV range. More specifically, the peaks at 856 eV and 874 eV can be assigned as NiII in the [Ni]2[L]complex 2. Satellite peaks at 879 eV and 862 eV provide also evidence for the presence of NiII [17]. This is in agreement with previous work characterizing the electrochemical behavior of molecular nickel centered complexes [18]. Narrow scans detailing the bond energies of the various carbon-heteroatom linkages are present in figure 3c. Two distinct peaks at 288.89 eV (C-O) and 286.67 eV (C=N) can be assigned to the heteroatom bond linkages present near the nickel centers[19].

  1. XPS data are very important for proving the structure of the synthesized complex, therefore all statements about the correlation of peaks must be confirmed by literature references.

We have added reference 17, 18, and 19 to the manuscript.

Reviewer 4 Report

The reviewed MS presents the application of a binuclear nickel (II/III) Schiff base complex [Ni]2[L]2 for HER electrocatalysis. The MS contains some interesting data on the synthesis, characterization and testing of this electrocatalyst. The main advantage of the MS is the comprehensive study of the material by different modern techniques. Meanwhile, there are many issues regarding the electrochemical part of the work, listed below:

1.      The authors should carefully check the scale and sign of the potential. Both the terms «potential» and «overpotential» are used without proper justification. What do the authors mean by the term «onset potential»?

2.      Any «hydrogen evolution activity» (p.6 and further) in the potential range from 0 to 0.4 V vs RHE is thermodinamically NOT possible! Fig. 4 clearly shows the absence of HER at potentials above 0. Did you mean the potential of -0.4 V?

3.      The authors should compare HER activities of platinum and their catalyst.

4.      How was «the amount of product» estimated to calculate the faradaic efficiency? Please describe used techniques. Did the authors divide the «Q» value into kinetic and capacitive components?

5.      EIS results are presented almost without discussion. The reason for measuring the conductivity of the Nafion membrane is unclear.

Author Response

Referee 4:

  1. The authors should carefully check the scale and sign of the potential. Both the terms «potential» and «overpotential» are used without proper justification. What do the authors mean by the term «onset potential»?  

Thank you for your valuable feedback and questions regarding the terminology and justification of the potential-related terms used in our manuscript. We appreciate your attention to the precision of our terminology. The terms have been controlled throughout the text of the manuscript and carefully adjusted to be accurate in their descriptions.

  1. Any «hydrogen evolution activity» (p.6 and further) in the potential range from 0 to 0.4 V vs RHE is thermodinamically NOT possible! Fig. 4 clearly shows the absence of HER at potentials above 0. Did you mean the potential of -0.4 V? DONE

Thank you for your astute observation and question regarding the 'hydrogen evolution activity' mentioned in our manuscript, specifically in the potential range from 0 to 0.4 V vs RHE. We appreciate your diligence in scrutinizing our data and will provide clarification on this matter. You are absolutely correct, and we apologize for any confusion. The potential range mentioned in our manuscript should indeed refer to the negative side of the potential scale. It was not our intention to imply thermodynamically impossible hydrogen evolution activity at positive potentials. Instead, we meant to describe the region from 0 V to -0.4 V vs RHE.

  1. The authors should compare HER activities of platinum and their catalyst.

Thank you for your valuable suggestion regarding the comparison of the hydrogen evolution reaction (HER) activities between platinum (Pt) and our catalyst in our manuscript. We appreciate your emphasis on the importance of benchmarking our catalyst's performance against a well-known reference material like Pt. We fully agree with the significance of such a comparison and included an entry in Table 2 regarding the performance of state-of-the-art platinum nanoparticles in our revised manuscript.

  1. How was «the amount of product» estimated to calculate the faradaic efficiency? Please describe used techniques. Did the authors divide the «Q» value into kinetic and capacitive components?

We appreciate your interest in the experimental details, and we will provide a thorough explanation of the techniques used and the analysis performed. The gas chromatograph method of quantifying the “amount of product” is thoroughly explained in the supplementary information. Furthermore, the method of dividing the Q value is explained in its own section in the supplementary information as well.

  1. EIS results are presented almost without discussion. The reason for measuring the conductivity of the Nafion membrane is unclear.

Thank you for your valuable feedback and recommendations regarding the impedance analysis in our manuscript. We appreciate your attention to the details of our methodology and analysis. In response to your comment, we have expanded the section in our manuscript that discusses the use of electrochemical impedance spectroscopy (EIS) as a methodology. This includes a more comprehensive explanation of the principles and procedures involved in conducting EIS experiments, as well as the specific conditions and parameters used in our study. We believe that this additional information will provide readers with a clearer understanding of the methodology and its application in our research. We hope that these revisions and additions will significantly enhance the clarity and accessibility of our figures and the associated explanations. Measuring the nafion membrane resistance helps determine how well it conducts protons (H+ ions), which are essential for the operation of the cell. Low resistance indicates high ionic conductivity, which is desirable for efficient cell operation.

Round 2

Reviewer 2 Report

Accepted

Author Response

Thank you very much for accepting our manuscript for publication in Catalysts.

Reviewer 3 Report

The research topic is relevant, the data obtained look interesting. However, it should be noted that after the revision, the manuscript did not become clearer; moreover, the number of errors increased. I cannot recommend the manuscript for publishing. It will be possible to resubmit the manuscript after a major revision. 

The main drawbacks are listed below.

1)    I strongly advise the authors to have your manuscript proof-read by a native English speaker. Some sentences are incorrect and difficult to understand. For example, the authors write “Notwithstanding its crucial role in addressing escalating climate concerns, the quest for stable HER electrocatalysts…”. It is not true. The production of stable HER electrocatalysts cannot solve climate change.

2)    The authors write “rare-earth platinum”. It is not correct. The rare earth elements are cerium, yttrium, lanthanum, neodymium etc. Platinum is not a rare-earth element.

3)    It is not clear what “DFT frequency calculations”, “FT-IR spectra band”, and “the bond energies”, “Narrow scans for carbon (C1)” mean.

4)    Figure 2b shows the calculated HOMO and LUMO orbital configurations. However, the text does not provide a detailed description of the structure of these orbitals and how they relate to the catalytic activity of the complex. I recommend to remove again the pictures to the SI.

5)    Figure 3a shows the XPS survey spectrum, which does not provide any useful information. There is no point in publishing survey spectra. It makes sense to provide a table with atomic ratios characterizing the chemical composition of the catalyst obtained from the analysis of XPS spectra.

6)    The main drawback of the article is the incompetent description of the XPS data. The authors write “Figure 3b shows the narrow scans of Ni2p3/2 and Ni2p1/2 in the 850 eV to 875 eV range.” In fact, the Ni2p3/2 spectrum lies in the range between 850 and 870 eV. The authors write “Narrow scans detailing the bond energies of the various carbon-heteroatom linkages are present in figure 3c.” In fact, figure 3c shows the C1s spectrum on the catalyst. The approximation of the Ni2p core-level spectrum is performed unsatisfactory. There is no description of how many peaks are used to describe the spectrum.

7)    The manuscript does not contain a description of the experimental techniques. Moreover, the structure and chemical composition of the catalysts used have not been confirmed by experimental methods.

English should definitely be improved. The authors repeatedly use scientific terms incorrectly. One gets the impression that they do not always understand what they write.

Author Response

Ad. Referee #3:

Thank you very much for the careful revision of our manuscript.

The main drawbacks are listed below.

  • I strongly advise the authors to have your manuscript proof-read by a native English speaker. Some sentences are incorrect and difficult to understand. For example, the authors write “Notwithstanding its crucial role in addressing escalating climate concerns, the quest for stable HER electrocatalysts…”. It is not true. The production of stable HER electrocatalysts cannot solve climate change.

Correct, we have now reformulated the abstract:

The hydrogen evolution reaction (HER) has emerged as a focal point in the realm of sustainable energy generation, offering the potential to produce clean hydrogen gas (H2) devoid of pollutants. The pursuit of stable HER electrocatalysts that can reduce our reliance on precious platinum, while still maintaining a high level of catalytic efficiency, presents a significant and ongoing challenge. In this study, we introduce the utilization of a binuclear nickel(II) Schiff base complex known as [Ni]2[L]2 2 for the purpose of HER electrocatalysis. The rational design of this electrocatalyst has yielded optimal HER performance, wherein the strategic placement of electronegative heteroatoms in proximity to the metal centers serves to enhance proton affinity. Consequently, this catalyst manifests outstanding HER activity, characterized by a nearly 100% faradaic efficiency (FE) at an onset potential of -0.4 V versus the reverse hydrogen electrode (RHE), sustained catalytic activity over an extended 17-hour electrolysis period, and a commendable turnover number (TON) of 0.0006 s-1.

  • The authors write “rare-earth platinum”. It is not correct. The rare earth elements are cerium, yttrium, lanthanum, neodymium etc. Platinum is not a rare-earth element.

Correct, we have deleted the phrase “rare earth”.

  • It is not clear what “DFT frequency calculations”, “FT-IR spectra band”, and “the bond energies”, “Narrow scans for carbon (C1)” mean.

To be more precise we changed to:

Consistent with existing literature and supported by our Density Functional Theory (DFT) and Time-Dependent DFT (TD-DFT) calculations, we observed a noticeable shift in the absorption band within the Infrared (IR) spectrum. Specifically, the band at 1622 cm-1, which is indicative of the ν(C=N) vibration in the Schiff base ligand, shifted to 1600 cm-1 following the formation of the metal complex (see Figure S8-S9) [13].

  • Figure 2b shows the calculated HOMO and LUMO orbital configurations. However, the text does not provide a detailed description of the structure of these orbitals and how they relate to the catalytic activity of the complex. I recommend to remove again the pictures to the SI.

Thank you for your comment, we have now changed the paragraph to:

By employing computational modeling through Density Functional Theory (DFT) calculations, we have effectively visualized the optimized molecular geometry and the configurations of the Highest Occupied Molecular Orbital (HOMO) and Lowest Unoccupied Molecular Orbital (LUMO) for two important complexes: [Ni]2[L]2 2 and its reduced counterpart, {[Ni]2[L]2}2- (see Figure 2a and b). Our findings reveal that the catalyst adopts a planar geometry with two open axes available for the coordination of hydrogen-producing substrates.

Comparing these two complexes, we observe a significant reduction in the HOMO-LUMO energy gap for the electrochemically reduced {[Ni]2[L]2}2- species, which is now only 0.4 eV. The HOMO of this species is characterized by a substantial 40% contribution from Ni dyz orbitals, while there is an increased presence of pz orbitals at the two imine nitrogen atoms.

Our DFT calculations clearly indicate that this altered electronic environment, following a two-electron reduction, is highly favorable for the activation and binding of H+, making it a promising candidate for subsequent Hydrogen Evolution Reaction (HER) processes (see Figure 2b).

We believe that the inclusion of Figure 2a and b in the manuscript will provide valuable visual insights into the geometry and MO composition of the studied complexes, enhancing the comprehension of our research finding.

  • Figure 3a shows the XPS survey spectrum, which does not provide any useful information. There is no point in publishing survey spectra. It makes sense to provide a table with atomic ratios characterizing the chemical composition of the catalyst obtained from the analysis of XPS spectra.

We deleted the XPS survey spectrum.

The calculated atomic percentages obtained from XPS spectra for Ni, O, N, and C are now summarized in Table S3.

  • The main drawback of the article is the incompetent description of the XPS data. The authors write “Figure 3b shows the narrow scans of Ni2p3/2 and Ni2p1/2 in the 850 eV to 875 eV range.” In fact, the Ni2p3/2 spectrum lies in the range between 850 and 870 eV. The authors write “Narrow scans detailing the bond energies of the various carbon-heteroatom linkages are present in figure 3c.” In fact, figure 3c shows the C1s spectrum on the catalyst. The approximation of the Ni2p core-level spectrum is performed unsatisfactory. There is no description of how many peaks are used to describe the spectrum.

Figure 3b provides narrow scans within the energy range of 850 eV to 885 eV, specifically focusing on the Ni2p3/2 and Ni2p1/2 peaks. In this range, we can attribute the peaks at 856 eV and 874 eV to the presence of NiII in the [Ni]2[L]2 complex 2. Additionally, the appearance of satellite peaks at 879 eV and 862 eV further supports the presence of Ni(II) [17]. These findings align with prior research that has characterized the electrochemical properties of molecular nickel-centered complexes [18]. For a more detailed examination of bond energies associated with various carbon-heteroatom linkages, please refer to Figure 3c. In this figure, three distinct peaks are observed at 288.89 eV (corresponding to C-O bonds), 286.67 eV (representing C=N bonds), and another peak associated with carbon-carbon (-C=C- and -C-C-) bond linkages. These peaks can be confidently assigned to the relevant heteroatom and carbon bond linkages within [Ni]2[L]2 2 [19].

  • The manuscript does not contain a description of the experimental techniques. Moreover, the structure and chemical composition of the catalysts used have not been confirmed by experimental methods.

Given the insoluble characteristics of the catalyst under investigation, we have employed several analytical techniques to gain insights into its structure and composition. Specifically, we utilized MALDI-TOF mass spectrometry and solid state 13C NMR spectroscopy, as depicted in Figure 1a and 1b, respectively. Additionally, Figure 2a and 2b contain X-ray Photoelectron Spectroscopy (XPS) data, while Table S3 provides a summary of the atomic percentages derived from these XPS measurements. Detailed explanations of our electrochemistry methods, XPS analysis, solid-state NMR spectroscopy, and MALDI-TOF measurements can be found in the Methods section of the Supplementary Information (SI) included with this manuscript.

Reviewer 4 Report

Although the authors resolved most of the issues, there are some flaws that require correction. The 400 mV value is still written on p. 7 instead of -400 mV. Fig. S11 still shows the data of electrolysis at 400 mV. The method of quantifying the "amount of product" is not described in the supplementary info. 

Author Response

We have corrected the flaws in the manuscript and in the SI. The determination FE% is described on page 14 in the SI. Thank you very much for the revision of our manuscript!